# Advancing Environmental Justice through the Integration of Traditional Ecological Knowledge into Environmental Policy

**Jennifer B. Rasmussen**

School of Public Health and Health Sciences, University of Massachusetts, Amherst, MA 01003, USA; jenniferbrown.rasmussen@gmail.com

**Abstract:** As our planet faces more frequent and severe environmental threats due to climate change (including threats to biodiversity), environmental justice will be essential to ensure that the costs and burdens of combating these threats are shared equally, borne by all people worldwide in a fair and equitable manner. If the past is any indicator, however, environmental problems—and their "solutions"—disproportionately affect poor communities and communities of color, including Indigenous communities. Despite these past injustices, Indigenous lands, which make up only 20 percent of the Earth's territory, contain 80 percent of the world's remaining biodiversity—evidence that Indigenous peoples are among the most effective stewards of the environment. A primary reason for this remarkable statistic is the use and practice of Indigenous Traditional Ecological Knowledge; ecological wisdom which has been passed down for generations and has been shown to strengthen community resilience in response to the multiple stressors of global environmental change. While the United States government has been slow to acknowledge the value of Traditional Ecological Knowledge, it has recently begun to incorporate that knowledge into environmental policy in response to the worldwide climate crisis. Continuing the integration of Traditional Ecological Knowledge into government environmental policy will ensure that such policies will be more effective at the federal, state, and local levels and more equitable in their application. Western scientists, government officials, and global leaders need to build trusting and co-equal relationships with Indigenous communities by actively listening to all cultures and respecting the many kinds of knowledge systems required to conserve the natural world and all living beings. This paper will address how incorporating Traditional Ecological Knowledge into U.S. policy would help safeguard the environment from further biodiversity loss and other ecological destruction, and advance environmental justice to ensure the fair treatment of all.

**Keywords:** planetary health; traditional ecological knowledge; environmental health; environmental justice; indigenous; policy

## 1. Introduction

Earth's ecosystems have existed and evolved for millions of years, resulting in diverse and complex biological communities living in balance with their environment. Everything that makes up the environment—plants, animals, insects, bacteria, fungi, and other organisms—is integral to the overall health and well-being of all inhabitants of this planet. First Nations people in Canada believe that every being has a purpose, deserves to be respected and cared for, and has a significant role to play in life. This powerful outlook embraces the idea that all people are connected to their communities, ancestors, the lands on which they live, future generations, and all the animals and plants that reside on their lands [1].

Earth is currently experiencing what many scientists call a sixth mass extinction. While the planet has experienced other mass extinctions in the past (typically caused by catastrophic events such as asteroid collisions), this is the first time that such an event is being driven by ecosystem-destroying, human-induced changes, such as fossil fuel

emissions, deforestation practices, and changes in land use [2]. Unless these practices cease or are drastically reduced, climate change and other environmental destruction will continue to affect the habitats of numerous species, leading many to extinction.

Protecting all lifeforms on Earth is crucial. As the planet faces more frequent and severe threats due to climate change, environmental justice will be critical to ensure that all people worldwide are receiving adequate care and attention. It is an unfortunate and infuriating truth that climate change disproportionately affects low-income communities and communities of color, including Indigenous communities, who experience climate change and other ecological disasters at a greater intensity when compared to non-Indigenous populations [3]. While the world addresses these environmental crises, greater attention needs to be put on socially and economically disadvantaged communities as they face the greatest risks based on where they live, their health, income, language barriers, and lack of resources. Areas with healthy ecosystems, in contrast, will be more resilient to climate change and can more adequately maintain the supply of ecosystem services which is essential to the health and well-being of all life forms.

Biodiversity, the term used to describe the variety of all life on Earth, maintains the health and resiliency of nature. The more abundant the biodiversity, the more secure all life is. Ecosystems weakened by biodiversity loss are not as capable of fulfilling their role as ecological life support, especially with environmental degradation and overpopulation [4]. Indigenous lands make up a small percentage of the Earth's territory yet contain 80 percent of the world's remaining biodiversity—evidence that Indigenous peoples are among the most effective stewards of the environment [5]. Among the reasons they are able to continue safeguarding some of the most biodiverse areas on the planet is their continued use and practice of Traditional Ecological Knowledge—a knowledge that has been passed down for generations in many Indigenous communities [6].

Implementing Traditional Ecological Knowledge has been shown to strengthen community resilience to the multiple stressors of global environmental change. While Western civilization is currently employing adaptation strategies to combat climate change and other environmental crises, they may actually create more adversities for Indigenous peoples, who are more effective in conserving the environment [6]. One adaptation solution would be to integrate Traditional Ecological Knowledge with Western science. This holistic approach would be advantageous for all, as it would blend ancient wisdom with policies and infrastructure aimed to protect the planet from worsening effects. However, in order to utilize Traditional Ecological Knowledge—either by itself or in conjunction with Western methods—Indigenous communities need to be protected and their effective ecological practices preserved.

Incorporating Traditional Ecological Knowledge into environmental policies also ensures that U.S. governmental action is effective at the federal, state, and local levels in a more just and equitable manner. Since taking office in 2021, the Biden administration has taken steps towards addressing these environmental challenges by implementing emission reductions, promoting clean energy, sustainable infrastructure, and environmental conservation. In doing so, the administration has acknowledged Indigenous-led conservation efforts and included Traditional Ecological Knowledge into their policy proposals—but more needs to be done. Incorporating Traditional Ecological Knowledge into U.S. policy would not only help safeguard the environment from further biodiversity loss and other ecological destruction, but advance environmental justice to ensure the fair treatment of all.

## 2. Environmental Justice

The United States Environmental Protection Agency (EPA) defines environmental justice as "the fair treatment and meaningful involvement of all people regardless of race, color, national origin, or income with respect to the development, implementation and enforcement of environmental laws, regulations, and policies" [7]. The concept of environmental justice is designed to address the incontrovertible truth that vulnerable communities bear a disproportionate impact of the pollution and contamination that stems from environmental

crises such as climate change, deforestation, and air pollution [8]. For example, many of the 574 federally recognized Tribes in the United States are disproportionately impacted by climate change, from drought and wildfires afflicting tribes in the Southwest to coastal Tribes (e.g., Alaskan Native communities, Quileute Nation in Washington, and the Seminole Tribe in Florida) grappling with erosion, permafrost thawing, flooding, and sea level rise [9]. As these crises directly affect the social and environmental determinants of health, including clean air, adequate food supply, safe drinking water, and safe shelter, the application of environmental justice to all communities is critical to avoid a disproportionate impact on those most vulnerable.

Supported primarily by communities of color (e.g., African Americans, Latinos, Asians, Pacific Islanders, and Native Americans), the environmental justice movement advocates that all people and communities receive fair and equal protection from environmental health hazards. It is axiomatic that the people living, working, and playing in the most polluted environments are typically people of color and those who are living in poverty. Indeed, research has shown that this is no accident, as these communities are "routinely targeted to host facilities that have negative environmental impacts—say, a landfill, dirty industrial plant or truck depot" [10]. For example, a 2020 report by the Shriver Center on Poverty Law found that 70 percent of Superfund sites (i.e., contaminated areas with hazardous waste left out in the open and improperly managed) in the United States are located within a mile of government-assisted housing. Residents living near these sites are predominantly people of color, children, the elderly, and disabled people [11].

In 1991, representatives for the People of Color Environmental Leadership Summit in Washington, DC drafted and adopted 17 Principles of Environmental Justice to be used as a defining document for the then-growing grassroots movement for environmental justice. According to these Leadership Summit representatives, the purpose of these principles is to build a global movement of all people of color to fight the destruction and taking of lands, and to [12]:

- Re-establish our spiritual interdependence to the sacredness of our Mother Earth.
- Respect and celebrate each of our cultures, languages, and beliefs about the natural world and our roles in healing ourselves.
- Ensure environmental justice; to promote economic alternatives which would contribute to the development of environmentally safe livelihoods.
- Secure our political, economic, and cultural liberation that has been denied for over 500 years of colonization and oppression, resulting in the poisoning of our communities and land and the genocide of our peoples.

At the time this summit took place, many existing white-led environmental organizations focused on conservation efforts. However, the attendees of this summit, coming from Black and Brown communities unjustly burdened by pollution, embraced the concept of environmental justice—the concept that all people are entitled to healthy environments where they work and live [13]. These principles highlight how all individuals have the right to be protected from environmental hazards while living in and enjoying clean and healthful environments.

Recognizing and protecting the natural world is another key element of environmental justice. As noted in one of the 17 Environmental Justice principles from the 1991 Leadership Summit, "environmental justice affirms the sacredness of Mother Earth, ecological unity and the interdependence of all species, and the right to be free from ecological destruction" [12]. This principle emphasizes that biodiversity conservation is essential for the health and well-being of all life forms. To ensure that communities and the environment are equally protected, it is critical that people, industries, and governments—particularly those with policymaking influence and ability—recognize the interdependent connection between humans and the environment.

Indigenous communities often understand this critical linkage, which has made their involvement in the environmental justice movement even more resonant. For example, in Canada the Indigenous Environmental Justice Project, an initiative based out of York

University, is working to remedy the gaps in environmental justice pertaining to Canadian Indigenous peoples who arguably face most of the existing ecological challenges. Considering Indigenous knowledge, principles, and values while also supporting these communities, providing necessary resources, and creating opportunities are critical steps towards equity.

Social inequalities across race, gender, and class can influence how the effects of biodiversity loss are felt and can worsen environmental degradation by unfairly driving poor communities (including those in developing countries) to pursue ecologically damaging survival and development strategies [14]. For example, Borneo (the largest island in Asia), suffers from one of the highest rates of deforestation in the world. Despite the presence of several protected national parks, forests continue to be unsustainably cut down for timber or cleared to make room for farms or plantations [15]. Forest clearing is one of the only well-paying employment opportunities in Borneo and workers often rely on these logging jobs to feed their families.

Another example of such an injustice is the maltreatment and murder of Indigenous environmental defenders in the Global South. These defenders risk their lives to protect governments, businesses, and non-state actors from overexploiting their lands. According to a recent Global Witness Report, "three people are killed every week while trying to protect their land, their environment, from extractive forces" [16]. More than 75 percent of the recorded attacks occurred in Latin America, while 78 percent of the total attacks in Brazil, Peru, and Venezuela took place in the Amazon [16]. Despite these atrocities, Indigenous defenders are still not receiving legal protection from their respective governments.

Indeed, the battle to fight climate change, environmental justice, and biodiversity loss are intertwined. Social and environmental justice should be central in addressing these ongoing crises as all individuals should be able to interact confidently with a safe, nurturing, and protective environment. Environmental justice can only be achieved when both cultural and biological diversity (i.e., biodiversity) is respected, protected, and celebrated.

## 3. The Importance of Biodiversity

Biodiversity is the biological variety and variability of living species on Earth. From plants, animals, fungi, and bacteria, biodiversity includes the ecological, evolutionary, and cultural processes that support life. It refers to the different levels of species and creatures that have interacted with the physical environment and have made Earth habitable for billions of years. These species and organisms work synergistically in biological communities called ecosystems to protect and support the ecological balance of all life [17].

Biodiversity is indispensable to the health and sustainability of all species. It makes ecosystems more resilient and is necessary for physical, mental, emotional, and spiritual health, as it plays a role in maintaining a healthy and diverse planet. Biodiversity also "shapes who we are, our relationships to each other, and social norms" [18]. These relational values are part of peoples' individual or collective sense of well-being, responsibility for, and connection with the environment [18].

Environmental destruction ruins habitats, releases sequestered carbon, contributes to biodiversity loss, and exposes people to viruses they had no previous contact with and for which they have no natural immunity. In contrast, biodiversity conservation reduces the risk of such zoonotic diseases as it provides more habitats for species and reduces possible contact between wild animals and humans. Having a rich blend of sources of biodiversity is a key element for preservation and protection.

Areas where the rate of biodiversity decline is the slowest are usually found on lands governed by Indigenous peoples [5]. Considered to be dedicated stewards of their natural environments, many Indigenous communities around the world have lived in harmony with their lands for generations [19]. Using and conserving natural resources while maintaining thriving ecosystems is a testament to their skills as sustainability practitioners. Therefore, it would serve humanity well to invest in biodiversity conservation and harness

Traditional Ecological Knowledge as the world is faced with the challenges of adapting to a changing climate caused by environmental degradation.

## 4. Traditional Ecological Knowledge

Traditional Ecological Knowledge is the "accumulation of knowledge, practice, and belief about relationships between living beings in a specific ecosystem that is acquired by Indigenous people over hundreds or thousands of years through direct contact with the environment, handed down through generations, and used for life-sustaining ways" [20]. This knowledge includes the understandings, skills, and principles of Indigenous peoples acquired through multigenerational histories of dealings with the natural world and adapting to variable ecological and social conditions, including colonization and globalization [21].

Many Indigenous communities around the globe have a deep understanding of how to sustainably manage and preserve natural resources. For generations, many of these communities have acted as stewards of the Earth, contributing to biodiversity conservation and ecosystem health. In fact, while Indigenous lands make up a mere 20 percent of the Earth's territory they contain 80 percent of the world's remaining biodiversity, which means that Indigenous peoples and local communities conserve far more of the Earth than National Parks and protected forests [5]. It is because Indigenous communities utilize Traditional Ecological Knowledge that they are able to effectively protect their territories.

Indeed, many Indigenous communities are vital to the different ecosystems that inhabit their territories and may, therefore, help enhance their resilience [22]. For example, the eastern coast of Mindanao in the Philippines is home to the Manobo people, who have occupied a region of that area known as Pangasananan. The Manobo people have relied on this land to cultivate crops, hunt, fish, and gather herbs and have implemented techniques to conserve the land for centuries [23]. It is thanks to these conservation practices that the Pangasananan region has remained ecologically intact.

Many Indigenous peoples do not consider themselves separate from nature but part of nature, and they acknowledge this unity of reverence and reciprocity with the Earth. Aaron Payment, a member of the Sault Tribe of the Chippewa Indians in Michigan, put it succinctly: "we see ourselves as part of [nature] because it's life-sustaining... our very lives depend on living in ecological balance with our natural resources" [23]. The living world is understood through this lens not as exploitable resources, but as a set of relationships and responsibilities. Robin Wall Kimmerer, a scientist, author, and member of the Citizen Potawatomi Nation, explained that it was—and continues to be—through the "actions of reciprocity, the give and take with the land, that the original immigrant became Indigenous. For all of us, becoming Indigenous to a place means living as if your children's future mattered, to take care of the land as if our lives, both material and spiritual, depended on it" [24]. Reciprocity is the notion that for the Earth to remain balanced and for the gifts Earth gives us to continuously flow, one must give back in equal measure for what one takes.

Traditional Ecological Knowledge is also valuable for identifying changes to biodiversity loss due to climate change. Many Indigenous communities have responded to the substantial impacts that climate change has had on their livelihoods through sustainable management for terrestrial and aquatic ecosystems. Various types of Indigenous tools and methods have helped communities adapt to climate change, while also promoting socio-ecological resilience [25]. For example, Indigenous peoples in Australia experience both fast and slow-onset climate change, with impacts already being felt and predicted to become more severe, including changes to social–ecological landscapes that will adversely impact long-standing culture and land-use practices. The exposure these Indigenous Australians have to the adverse impacts of climate change is compounded by existing socioeconomic disadvantages such as inadequate health and educational services, insufficient infrastructure, and limited employment opportunities—all of which are linked to colonial and post-colonial periods [26].

Collectively, Indigenous communities should be considered the world's biggest conservationists, though they rarely receive that recognition. Despite Traditional Ecological Knowledge existing for centuries, many Indigenous groups and their wisdom have been discounted. A member of the Kashia Pomo Tribe in California observed that "tribal knowledge in land conservation is often used and quoted but rarely matters until a white person says it. Unfortunately, that's just the truth of the last 100 years of land conservation in the U.S." [23]. Though there has been somewhat of a shift in recent years in which there is greater recognition and support for the tremendous contributions of Indigenous peoples, there is still disagreement as to who can best manage the most effective conservation practices: Indigenous communities or Western civilization.

As many Indigenous peoples have long understood and practiced, Traditional Ecological Knowledge has, at its core, a belief that all life should be sustained. They view both themselves and nature as part of an extended ecological family. Protecting their ways of life is a critical environmental justice issue as they have often suffered due to the continued dominance of Western views. Historically, environmental justice has been a significant issue for Native Americans in the United States. One such example of an infamous environmental injustice was the development of the uranium mining operations on the Navajo Nation in the 1940s. Before uranium was publicly acknowledged as a carcinogenic contaminant, over 500 uranium mines had been established and then abandoned on Navajo Nation, with human health and ecological damage left in its wake. Uranium mining disproportionately affected these poor and disadvantaged communities, as they targeted Indigenous peoples struggling to find work, introducing them to hazardous exposure to uranium in the process [27].

It is unfortunate that environmental policy has often been created about issues facing Native Americans without their input [27]. Instead, Native Americans should have the opportunity to address environmental problems using their Traditional Ecological Knowledge, which has ties to natural laws respected by tribal communities and provides the foundation for addressing the complex relationship between nature and humans. Furthermore, policies addressing environmental concerns that non-Native American stakeholders develop can have unintended adverse effects on Native American communities and, therefore, cause more harm than good [27]. The more that these policies can incorporate Indigenous peoples' perspectives, the more progress can be made to attain environmental justice.

## 5. Adaptation and Policy

### 5.1. What Has Been Done and What Is Working?

As a way of synergistically blending traditional and modern environmental conservation practices, some Traditional Ecological Knowledge is already being applied to current environmental policies with some success. In fact, the coordination of Traditional Ecological Knowledge and current climate policy has been on the upswing. In 2014, a subcommittee of Indigenous scholars developed "Guidelines for Considering Traditional Knowledge in Climate Change Initiatives" that laid the groundwork for integrating Traditional Ecological Knowledge into federal agency policies. More specifically, these guidelines would improve the understanding of how Traditional Ecological Knowledge would be incorporated into climate initiatives, provide guidance to those engaging in efforts that contain Traditional Ecological Knowledge, and increase mutually beneficial interactions between tribal and non-tribal partners [28].

In November 2021, the White House hosted the Tribal Nations Summit, in which the Biden administration published a memorandum to recognize Traditional Ecological Knowledge as "one of the important bodies of knowledge that contributes to the scientific, technical, social, and economic advancements of our nation" [9]. The administration also pledged to give direction to federal agencies on how Indigenous wisdom can amplify empirical science and decision making across the federal government. For Tribal leaders this was a significant development, as the Biden administration thoughtfully and substan-

tially acknowledged the importance of Traditional Ecological Knowledge fighting climate change [9].

As the Biden administration and others have made clear, the integration of Indigenous knowledge and Western science is not only possible, but desirable. For example, Confederated Tribes of Siletz Indian and Cherokee Tribes are taught ways to understand the natural world and interact with it to heighten awareness of their current surroundings and environment, akin to building a partnership. Indeed, acquiring, understanding, and maintaining Traditional Ecological Knowledge via adaptation and resilience allows for a holistic connection with nature. While some consider it "walking in two worlds" [9] many Indigenous communities and environmental policymakers understand the benefits. Even though Western science can be more taxonomic and separatist, there appears to be a shift towards the integration of these two disciplines and, in certain areas, the benefit of that coordinated policy is encouraging [9]. For example, due to worsening wildfires and floods and a massive loss of biodiversity, many Western researchers are now asking original land stewards (i.e., Indigenous peoples) for help restoring ecosystem balance. In Alaska, Inuit communities are "teaching researchers about biodiversity and sea ice conditions in the Arctic; in the American West, tribal members are collaborating with land managers to burn small fires to prevent big ones" [9].

Some Indigenous groups are incorporating their Traditional Ecological Knowledge with other technologies to cope with the impacts of climate change, because when adaptation strategies are grounded in local ecology and culture, effective climate solutions can be produced. For example, some Indigenous groups in North America are striving to cope with climate change by focusing on the economic opportunities that may increase demand for wind and solar power to make tribal lands an important resource for renewable energy, replacing fossil fuel and decreasing greenhouse gas emission [22]. Implementing such strategies as policy would protect the health of Indigenous communities from worsening climate impacts, while also encouraging greater prosperity and better well-being.

In April 2022, the United States Department of the Interior announced that Indigenous communities would receive $46 million in climate resilience funding. As part of President Biden's Bipartisan Infrastructure Law (which largely focuses on climate change mitigation), Indigenous communities will be given funds to support projects and initiatives to strengthen climate change resilience [29]. The Biden administration recognizes the environmental injustice that Indigenous peoples experience throughout the country, as they are disproportionately affected by climate change. As Secretary of the Interior, Deb Haaland, stated, "the effects of climate change continue to intensify [and] Indigenous communities are facing unique climate-related challenges that pose existential threats to tribal economies, infrastructure, lives, and livelihoods" [30]. For example, Alaska Natives are among the first climate refugees in the United States, as nearly 90 percent of their communities are threatened by flooding and erosion while also dealing with intense drought and extreme heat [30].

### 5.2. What Has Not Worked and What Needs to Be Improved?

U.S. policies, funding, and global calls to action have, thus far, been important. Measures taken towards elevating Indigenous Traditional Ecological Knowledge in federal scientific and policy processes are particularly substantial, as they are proven solutions against tackling environmental destruction caused or worsened by climate change. By having Traditional Ecological Knowledge as the foundation of global conservation actions and policies, we have a greater probability of achieving true environmental justice and further ecological consequences. Despite the positive steps taken by global and national agencies, it is simply not sufficient if we are to achieve global environmental goals.

Even with their levels of success, certain adaptation strategies have proven to be challenging, as they can have adverse cultural and environmental effects. For example, reindeer herders from the Saami community in northern Europe fear that the use of supplemental nutrition (due to climate-related food insecurity) could profoundly change

their cultural livelihood and cause reindeers to become tamer [31]. Another example is the impact of biofuel initiatives on certain areas. As Siham Drissi, Program Management Officer of Ecosystems at the United Nations Environment Programme (UNEP) explained, though biofuel initiatives aim to decrease greenhouse gas emissions, they may negatively affect the ecosystems, water supply, and landscapes in which Indigenous peoples depend. So, while they may lead to increased monoculture crops and plantations, there would be a subsequent decline in biodiversity, food, and water security [6].

A further concern among Indigenous communities is the potential adverse impact of expanded protection zones. The G7 (i.e., Group of Seven) is an intergovernmental organization of the world's largest developed economies: Canada, France, Germany, Italy, Japan, United Kingdom, and the United States. The G7 has a "30 by 30" goal in which they are committed to conserving 30 percent of their lands and waters by 2030. While this seems to be an admirable goal, many Indigenous communities are concerned, as they worry that reaching this goal could come at the expense of Indigenous land rights [23]. For example, a worst-case scenario is that the G7 countries and conservation organizations violate Indigenous land rights as they expand protected areas, especially as many of the remaining biodiversity hotspots are located on Indigenous lands.

The latest UN Intergovernmental Panel on Climate Change (IPCC) report from February 2022 issued a dire warning about the consequences of environmental inaction and stressed the urgency that protecting areas is a key climate adaptation measure. The authors of this report affirmed that safeguarding and restoring natural environments (e.g., wetlands, oceans, and forests) can lower the risks climate change poses to people, as well as support biodiversity, store carbon, and provide many other benefits for human health and well-being. Climate change is increasing the number of threats to people, such as flooding, droughts, wildfire, heatwaves, and rising sea levels. Such hazards can either be reduced or aggravated, depending on how land and water are managed and protected [32]. Furthermore, this report acknowledged that adaptation efforts benefit from the inclusion of Indigenous knowledge.

While there is cause for concern about the roadblocks experienced in attaining environmental justice and sustainability, solutions do exist, as evidenced by the significant advantages of implementing Traditional Ecological Knowledge into climate adaptation strategies and policy. However, more needs to be done. One suggestion for moving toward greater environmental justice is to acknowledge the tribal sovereignty of Indigenous communities. Tribal sovereignty refers to Indigenous peoples having rights to their land, resources, people, and sacred sites. "Pushing federal and state officials to seek the consent of tribes when building new mines, pipelines, highways, and other infrastructure that will impact tribal lands, sacred sites, and burial grounds... is key to empowering tribes to tackle climate change" [33]. By understanding that Indigenous peoples are true ecological stewards, this action would be advantageous and responsible, as it would allow them to adopt adaptation and mitigation strategies to defend against imminent environmental threats. Granting sovereignty would enable better opportunities for Traditional Ecological Knowledge implementation and protection of the natural world.

Another recommendation is to have more Indigenous leaders in positions of power. While there have been, and continue to be, influential Indigenous leaders, such as Secretary Deb Haaland (who became the first Native American to serve as a cabinet secretary with Biden's administration), representation is seriously lacking, and not just in politics, but in various organizations around the world. This is largely due to the historical marginalization and disregard of Indigenous voices dating back to early colonialism. Indigenous leadership can offer a wealth of underutilized conservation opportunities that past marginalization has denied to Indigenous communities and the entire country [34]. Having more representation in leadership positions would elevate the environmental justice movement, as it centers itself on ecological protection and safeguards human rights, especially for vulnerable communities.

A final suggestion would be to expand voting rights in the United States. Prioritizing Indigenous leadership requires making voting more accessible so that all citizens can participate in elections. Free and fair elections and the right to vote are the foundation of American democracy. However, for generations, "voters of color have faced discriminatory policies and other obstacles that disproportionally affect their communities. These voters remain more likely to face long lines at the polls and are disproportionately burdened by voter identification laws and limited opportunities to vote by mail. Limited access to language assistance remains a barrier for many voters" [35]. Executive departments and agencies need to partner with state, local, tribal, and territorial election officials to protect and encourage the right to vote, eliminate discrimination, and other obstacles to voting [35]. It is important to have representation in politics across all identities of race, sexual orientation, and socioeconomic status. A representative democracy needs to be representative of all the populations it serves.

## 6. Conclusions

The natural world—and all lifeforms on Earth—are under severe threat from numerous environmental crises. Biodiversity loss from habitat destruction and over-harvesting of natural resources has serious implications for the health of humans, animals, and their shared environment. The world is also starting to witness the extreme consequences of dangerous environmental changes at a more accelerated pace due to climate change. The release of harmful greenhouse gases, deforestation, and destructive agricultural habits have created a dangerously warm planet. As a result, in recent years climate change has affected the lives of all people, especially those who are more vulnerable and disadvantaged, including many of the world's Indigenous communities.

The environmental justice movement focuses on safeguarding these Indigenous peoples, as they are more prone to experience harm from environmental hazards. All beings deserve to live in healthy environments where they can live, grow, and thrive. However, many Indigenous communities (along with many other poverty-stricken communities and communities of color) often experience racial discrimination and are subsequently neglected (a problem that dates back to early colonialism). Despite centuries of bigotry, Indigenous peoples have been responsible for conserving most of the planet's thriving biodiversity.

Living in harmony with nature is a fundamental part of the values and beliefs of many Indigenous peoples, who have been devoted stewards of the Earth and have understood the connection between their health and the health of the planet. This interconnectivity is also heavily rooted in reciprocity so that the Earth can remain in balance and move towards a more sustainable future as a collaborative, collective society. Indigenous communities worldwide defend the Earth's biodiversity because they understand the importance of this interconnection and know that Earth needs to be protected for future generations.

Traditional Ecological Knowledge has been a part of Indigenous wisdom and culture for centuries. It has created the most sustainable and effective solutions in fighting against environmental degradation that has been caused, or been made worse by, climate change. Despite recent growing global recognition that Indigenous peoples are responsible for this invaluable Traditional Ecological Knowledge, more needs to be done to implement it into policy—and urgently. Time is running out before the Earth hits the acknowledged critical temperature level of 1.5 degrees Celsius above pre-industrial levels, indicating irreversible climate change. Western scientists, government officials, and global leaders need to build trusting and co-equal relationships with Indigenous communities by actively listening to one another and respecting the many kinds of knowledge systems required to conserve the natural world and all living beings. All hands need to be on deck; this is a global problem, and its resolution must be as immediate and far-reaching as the problem we face.

**Funding:** This research received no external funding.

**Institutional Review Board Statement:** Not applicable.

**Informed Consent Statement:** Not applicable.

**Data Availability Statement:** Not applicable.

**Conflicts of Interest:** The authors declare no conflict of interest.

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
