# Peer review of "Advancing Environmental Justice through the Integration of Traditional Ecological Knowledge into Environmental Policy"

_challenges, doi:10.3390/challe14010006_

Round 1

Reviewer 1 Report

The article is interesting, and worth publishing. However, I think more emphasis could be put on traditional ecological knowledge and its implementation. The part biodiversity (3) do not add up novelty to the issue. So I suggest rather to limit this part to one paragraph, and add more content on the article topic.

Author Response

Dear Reviewer,

Happy New Year! Thank you for taking the time to read my manuscript and provide constructive feedback. I have considered your helpful edits and made the changes you recommended. Please feel free to review my updated manuscript and let me know if you have any additional feedback. 

I greatly appreciate your time. Thank you again.

Take care,

Jennifer Rasmussen

Reviewer 2 Report

The reviewed article deals with the issue of the Advancing Environmental Justice Through the Integration of Traditional Ecological Knowledge into Environmental Policy . The layout of the work is clear, coherent and logical. The division of content and systematics adopted by the author does not raise objections. The research questions were formulated in a precise manner. Outlined research theses have been well proven. They were correctly verified and presented in a clear way in the conclusions. Research methods were also correctly applied. The article contains references to the most important publications concerning the title issue.

Author Response

(The authors gave the same response as above.)

Reviewer 3 Report

This paper explores the importance of including traditional ecological knowledge in environmental policy to advance environmental justice concerns. The piece is well-written, and the ideas and statements are highly relevant in the context of accelerated biodiversity loss. However, I would like to provide the following comments.

1.       Interestingly, the author has decided to refrain from engaging in more detail with the environmental justice framework and its three dimensions: recognition, distribution, and procedure. Because of the subject of the paper, an explanation of what recognitional and procedural justice means for incorporating this type of knowledge into environmental policy would better support the discussions and conclusions of this contribution.

2.       Some statements in the text need more discussion or a reference that supports them, so I suggest a careful second revision to identify missing references. Some examples that I found:

While the world addresses these environmental crises, greater attention needs to be put on socially and economically disadvantaged communities as they face the greatest risks based on where they live, their health, income, language barriers, and lack of resources (page 2, reference?)

While Western civilisation is currently employing adaptation strategies to combat climate change and other environmental crises, they may create more adversities for Indigenous peoples who are more effective in conserving the environment (page 2, here. I would also suggest looking at the literature on maladaptation)

Biodiversity also “shapes who we are, our relationships to each other, and social norms. (page 5, reference?)

Areas where the rate of biodiversity decline is the slowest are usually found on lands governed by Indigenous peoples (page 6, reference?) 

3. Section 3 could be substantially shortened to give more room for the paper's main keywords: environmental policy, environmental justice, and traditional ecological knowledge. 

4.       Cultural ecosystem services are not the only type of services supported by biodiversity (page 5). I suggest engaging more with the ecosystem services literature, particularly for regulating and provisioning services, if the author intends to connect the importance of biodiversity conservation to the benefits humans can obtain from ecosystems. In this line, I would also incorporate a definition of “ecosystem services” for readers unfamiliar with this term.

4.       Surprisingly, the Sustainable Development Goals are mentioned in this paper as one of the existing and working mechanisms for incorporating traditional ecological knowledge in environmental policy (page 8). Instead, the concept of sustainability has been criticised for being inherently in contrast to the traditional views of indigenous communities. This and related jargon have been mostly developed without considering the perspectives and realities of these communities to avoid contrasting with current and powerful agendas. This is a critical environmental justice issue because if indigenous voices and perspectives are excluded, communities also have fewer chances to participate actively in monitoring frameworks and decision-making processes.

5.       Even if the paper provides examples from different areas of the world, its focus is very centred on the context of the United States. This is ok but should be more explicitly stated in the title or the abstract. This is relevant because some of the statements and suggestions in section 5.2 cannot be generalised to other regions or countries (e.g., “a final suggestion would be to expand voting rights in the United States”).

6.       In relation to the last point,  the paper is missing some critical issues on the relationship between environmental policy, traditional knowledge, and environmental justice that are found outside the global north. For example, in particular areas of the world, such as Latin America, indigenous communities have been marginalised and excluded from legal protection by the State. As a result, they face severe violations and affectations due to the overexploitation of nature in their territories, which makes it impossible to incorporate traditional knowledge into a conservation perspective. Even if this issue does not need to be explored in detail, at least it should be mentioned, particularly after the publication of the last Global Witness Report that has emphasised how indigenous defenders are deprived of their freedom, threatened, and murdered, which aggravates environmental injustice and limits the development of communities. 

Author Response

Dear Reviewer,

Happy New Year! Thank you for taking the time to read my manuscript and provide constructive feedback. I have considered your helpful edits and made the changes you recommended. The one comment I was not able to address was regarding the environmental justice framework and its three dimensions. I had a little trouble completely understanding this concept and was unsure as to where it should be included in my manuscript. I also, unfortunately, ran out of time before meeting my deadline. Perhaps, if I get a second chance to update my manuscript, I will be able to look at this again. Please feel free to review my updated manuscript and let me know if you have any additional feedback. 

I greatly appreciate your time. Thank you again.

Take care,

Jennifer Rasmussen